# Maternal mental health matters: Indicators for perinatal mental health—A scoping review

Elly Layton[1], Alexandra Roddy Mitchell[1,2], Elissa Kennedy[1,3], Allisyn C. Moran[4], Francesca Palestra[4], Neerja Chowdhary[5], Shanon McNab[6], Caroline S. E. Homer[1]*

1 Maternal, Child and Adolescent Health Program, Burnet Institute, Melbourne, Australia, 2 Department of Obstetrics, Gynaecology and Newborn Health, University of Melbourne, Melbourne, Australia, 3 Murdoch Children's Research Institute, Melbourne, Australia, 4 Department of Maternal, Newborn, Child and Adolescent Health and Ageing, World Health Organization, Geneva, Switzerland, 5 Department of Mental Health and Substance Use, World Health Organization, Geneva, Switzerland, 6 MOMENTUM Country and Global Leadership, Jhpiego, Washington, District of Columbia, United States of America

* caroline.homer@burnet.edu.au

**Data Availability Statement:** All relevant data are within the manuscript and its Supporting information files.

## Abstract

Perinatal mental health disorders are a significant contributor to morbidity and mortality in childbearing women. The World Health Organization recommends all women be screened for mental health disorders postnatally and have diagnostic and management services available. There are, however, currently no global indicators in use which measure the status and progress of perinatal mental health. The aim of this scoping review was to identify existing perinatal mental health indicators and propose a core set which could be used at a global level. We used the Global Perinatal Mental Health Theory of Change as the conceptual framework. We found 25 indicators for PMH aligned with the Global Perinatal Mental Health Theory of Change, which were condensed to form a core set of nine indicators These core indicators include the proportion of women with depression, anxiety, post-traumatic stress disorder, psychosis, or adjustment disorders in the perinatal period; the proportion of women screened for these services; the proportion who have access to services following a positive diagnosis; and, the proportion of healthcare providers trained to provide mental health care. This review forms part of the foundational work for the development of a global monitoring framework which would be able to monitor progress towards the provision of universal high quality perinatal mental health care.

## Introduction

Perinatal mental health (PMH) disorders are among the most common morbidities of the perinatal period [1, 2]. The perinatal period represents a unique time in a woman's life when there are significant physical and social changes [3]. During this time, women are at higher risk of new onset of mental health disorders, or recurrence of pre-existing mental health conditions [4, 5]. PMH disorders are associated with morbidity and mortality [4]. In women, they are associated with increased obstetric complications, reduced quality of life, substance use, and suicide [6–8]. They increase the likelihood of women experiencing poverty, physical

**Funding:** CSEH and EK received a consultancy fee from WHO in Geneva to undertake this review. They worked in collaboration with WHO in relation to the study design, data collection and analysis, decision to publish and preparation of the manuscript. The WHO colleagues are co-authors of the paper.

**Competing interests:** The authors have declared that no competing interests exist.

health complications and intimate partner violence [8]. Further to this, PMH disorders can increase the risk of preterm birth, and poor fetal and infant growth and may impact the mother-infant bond [9, 10]. In children, caregiver PMH disorders can lead to poor neurodevelopmental outcomes and an increased risk of mental health disorders in adolescence [11, 12]. At the societal level, PMH disorders add a significant social and economic burden [4]. Globally, it is estimated that the prevalence of PMH disorders is 10–20% [10, 13], however, the majority of research has been performed in high-income countries [HIC], and rates are thought to be increased in low- and middle-income countries (LMIC) [11, 14].

The past two decades have seen a growing focus on the importance of PMH, however there are currently no global indicators that can measure the status or progress towards universal access to high-quality PMH care [10, 15, 16]. The prevention, identification, and treatment of women with PMH disorders is now included in multiple World Health Organization [WHO] guidelines for maternal and child health [16–18]. The Sustainable Development Goals also include the promotion of maternal mental health [19]. However, there is significant variation in policy, guidelines, service availability, screening tools, and implementation approaches worldwide [4, 9].

Indicators are instruments with the objectives of improving population health and reducing inequalities, and are a vital means of informing decision-making [20]. A monitoring framework with key indicators would enable consistent measurement of the status of PMH globally. It would inform actions aimed at improving the prevention, care, and treatment of PMH disorders, and facilitate monitoring and evaluation of progress towards specified targets. Therefore, the aim of this review is to identify existing PMH indicators published in the literature and create a core set of indicators to inform future development of a monitoring framework which can be used to measure PMH globally.

## Methods

A protocol including the scope and aims of this research as well as all subsequent findings was reviewed and refined in consultation with: WHO Department of Maternal, Newborn, Child, Adolescent Health and Ageing; WHO Mother and Newborn Information Tracking Outcomes and Results (MoNITOR) group; WHO Department of Mental Health and Substance Use; and members from the USAID MOMENTUM Country and Global Leadership PMH Community of Practice (S1 Appendix). Findings were reported using the Preferred Reporting Items for Systematic Reviews and Meta-Analyses Extension for Scoping Reviews (PRISMA-ScR) checklist (S1 Table) [21].

### Conceptual framework

The Global PMH Theory of Change (PMH ToC), supported by USAID's MOMENTUM Country and Global Leadership (MOMENTUM) project, was developed following MOMENTUM's landscape analysis on the burden of PMH disorders [22, 23]. The PMH ToC offers a framework to guide global PMH development towards the goal of widely accessible, high-quality PMH prevention, care and treatment services. It is based on a social ecological model that places the woman at the centre and incorporates interventions and outcomes across five domains: individual, interpersonal relationships, community, service delivery ecosystem and policy landscape. The impact includes improved PMH and wellbeing for all women [23]. The WHO Guide for Integration of Perinatal Mental Health includes methods of integrating perinatal mental health into routine maternal and child health services [24]. Based on this guideline and informed by findings of the scoping review, we adapted the PMH ToC to focus on individual outcomes and service delivery ecosystem interventions. This review only includes

indicators which align with the guideline and does not include risk factors for the development of PMH disorders as included in the PMH ToC.

## Search strategy and screening processes

The search strategy combined terms related to the perinatal period, mental health, and indicator (S2 Table). We performed the systematic search in three databases (MEDLINE, Embase and PsycINFO). Studies published between 1 January 2000 and 4 October 2024, were included. This time frame was selected to ensure contemporary studies were included. Studies were screened independently by two members of the research team (EL & ARM]) with any discrepancies on the inclusion/exclusion resolved by discussion. The process was managed using Covidence software [25]. We also extended our search to non-peer-reviewed material, thematically searching the websites of WHO, OECD, UNICEF Multiple Indicator Cluster Surveys (MICS), and The Demographic and Health Survey [DHS] for databases and reports which may include indicators for PMH [15, 26–28]. We also searched Governmental websites from English speaking countries which were referenced in peer reviewed sources and therefore likely to contain indicators (United Kingdom, Australia, United States of America). For each of these resources, available indicator dashboards were systemically reviewed for relevant indicators and the keyword 'perinatal mental health' was searched. Finally, we reached out to researchers working in this field for any additional relevant materials, which did not result in extra resources.

## Inclusion and exclusion criteria

The definition of PMH used in this review was taken from the landscape analysis and PMH ToC, which define PMH as "mental health during the perinatal period" [22]. Whilst the perinatal period is inconsistently defined within literature [22, 23], in this review, we defined the perinatal period as the time from the beginning of pregnancy to one year following the birth or end of the pregnancy [18]. The common mental health disorders which occur during the perinatal period include depression, anxiety and somatic disorders, while psychosis and substance use disorders occur more rarely [22]. This scoping review considered indicators from any setting which aligned with this definition, including both peer-reviewed journal articles and grey literature. After feedback from consultations with WHO and MoNITOR group members, we narrowed our scope and excluded indicators which measure risk factors and social determinants of PMH disorders, and indicators which measured mental health generally but were not specific the perinatal population. The quality of the indicator was not a criterion for inclusion. For the peer reviewed search, we did not limit by language of publication but used translation services for papers in a language other than English. The non-peer-reviewed material was searched in English only due to difficulty with navigation and translation of non-English webpages.

For inclusion, resources must have had:

1. An indicator which specifically measures perinatal mental health.

2. A definition of the indicator.

3. A publication date from 1 January 2000 to 4 October 2024.

The exclusion criteria were that the:

1. Indicator was not specific to perinatal population.

2. Indicator measured a risk factor and social determinant of PMH only.

3. Indicator had not yet been developed.

## Data extraction and analysis

Indicators were extracted individually from each resource. All available indicator data such as: indicator name, definition, numerator, denominator, and data source were compiled (S3 Table). Indicators with similar definitions were grouped together. Where it was determined that the indicators measured the same outcome, we proposed an indicator definition by combining them. (Table 1). The indicators in this core list were then mapped to the relevant level in the PMH ToC. The quality of each indicator was outside the scope of this review and hence was not assessed.

## Results

A total of 5808 resources from three databases were retrieved (Fig 1). An additional 12 resources from other sources including government websites and international agency websites were also retrieved. After duplicates were removed 4082 records were screened at the title/abstract stage and 109 at full text. Overall, 2 studies could not be retrieved and 91 were excluded at full text screening, resulting in 18 resources for inclusion.

From the 18 resources which were included, 25 indicators which measured PMH were extracted (S3 Table). Of these, 15 studies came from the database search of peer reviewed material with 17 indicators extracted. The grey literature search of webpages found the additional three resources, with eight indicators extracted. These sources were all from governmental websites (United Kingdom (UK) (n = 6), Australia (n = 1) and the United States of America (U.S.) (n = 1)]. There were no indicators found relating to PMH through searching the webpages of WHO, OECD, MICS or DHS [15, 26–28]. Of the 18 resources, 16 were from HICs (USA, UK, Australia, Ireland, Netherlands and Belgium]) and two were from LMICs (Mexico and Kenya).

The process of compiling indicators resulted in nine core indicators (Table 1). The proposed indicators were mapped to the PMH ToC, with six relating to individual measures of PMH, and three relating to measures of the PMH service delivery ecosystem.

## Discussion

The aim this scoping review was to identify existing perinatal mental health indicators and propose a core set which could be used at a global level. We found 25 indicators for PMH aligned with the PMH ToC, which were condensed to form a core set of nine indicators. This set of indicators can contribute to the development of a monitoring framework which if implemented can measure countries' progress towards widely accessible, high-quality provision of PMH care services.

We used the PMH ToC as the conceptual framework. Of the five domains of the ToC, we found PMH specific indicators for only two—individual and service delivery ecosystem. Broadly speaking, the included indicators in the individual domain aim to measure the prevalence of PMH disorders, while those identified in the service delivery ecosystem domain aim to measure service provision and quality. We were unable to identify any PMH specific indicators across the other three domains. Hence, many areas recognised as central to improving PMH may be without existing indicators. These, for example, include things such as understanding socio-cultural norms [9], community engagement with PMH promotion [10], the availability of treatment options; counselling and psychotherapy, accessibility of psychotropic medications and availability of inpatient care if required [4, 47], and policies that address PMH [48].

**Table 1. Core list of indicators mapped against PMH ToC.**

| Theory of change domain | Indicator short title | Proposed indicator | Indicator from resource |
|---|---|---|---|
| Individual | Postpartum depression | 1. Proportion of women with postpartum depression | Postpartum depression as assessed by the PHQ-2 or EPDS [29] |
| | | | The number of women who are identified as suffering from post-natal depression, divided by the total number of women [30] |
| | | | Percentage of women having a live birth who experienced depressive symptoms after pregnancy [31] |
| | | | Percentage of women who have recently given birth who reported experiencing postpartum depression following a live birth [32] |
| | | | The estimated number of women with postpartum depressive symptoms [33] |
| Individual | Perinatal depression and anxiety | 2. Proportion of women with depression or anxiety in perinatal period | The estimated number of women with mild-moderate depressive illness and anxiety [34] |
| | | | The estimated number of women with severe depressive illness [34] |
| | | | Patient-reported depression during antenatal and postnatal care periods [35] |
| Individual | Postpartum psychosis | 3. Proportion of women with postpartum psychosis | Acute psychosis among women during childbirth episode [36–38] |
| | | | The estimated number of women with postpartum psychosis [34] |
| Individual | Adjustment disorders | 4. Proportion of women with adjustment disorder and distress in perinatal period | The estimated number of women with adjustment disorders and distress [34] |
| Individual | Post-Traumatic Stress Disorder | 5. Proportion of women with Post-Traumatic Stress Disorder in perinatal period | The estimated number of women with post-traumatic stress disorder [34] |
| Individual | Mental health conditions | 6. Proportion of women with mental health conditions in perinatal period | The estimated number of women with chronic serious mental illness [34] |
| | | | Percentage of pregnant women with psychological or psychiatric problems [39, 40] |
| Service delivery ecosystem | Screening for mental health conditions | 7. Proportion of women screened for mental health conditions in perinatal period | Whether screening for mental health risk using a validated screening tool has been conducted during the antenatal period [41] |
| | | | Proportion of women who were screened for postpartum depression after a live birth [42] |
| | | | Documented evidence that the mother and her family/partner were encouraged to advise their public health nurse about mental health history, changes in mood, emotional state and behaviour that are outside of the mother's normal pattern. Care plan initiated as appropriate [43] |
| | | | Documented evidence that verbal and written information in relation to signs and symptoms of postnatal depression and preventative measures were given to the mother and partner if present. Mother advised to contact public health nurse service if symptoms occur. Care plan initiated as appropriate [43] |
| | | | Every pregnant woman is screened during the perinatal period at fixed moments for psychosocial vulnerability [44] |
| Service delivery ecosystem | Mental health services | 8. Proportion of women with access to mental health services during the perinatal period | Whether a woman in the perinatal period had access to mental health services since a disaster [45] |
| | | | Access to mental health services during antenatal care [46] |
| Service delivery ecosystem | Healthcare workers trained to manage psychological problems | 9. Proportion of healthcare providers trained to provide mental healthcare in perinatal period | Every health and social care provider has been expertly trained in dealing with psychological and social problems [44] |

Of the 25 indicators identified for PMH, only two were from papers from an LMIC [Kenya and Mexico], with the rest from high-income countries. Therefore, the majority of the indicators may not be applicable to LMICs. This has also been highlighted in the pilot study by Al-Shammari et al. [35] in Kenya which used the International Consortium of Health Outcomes

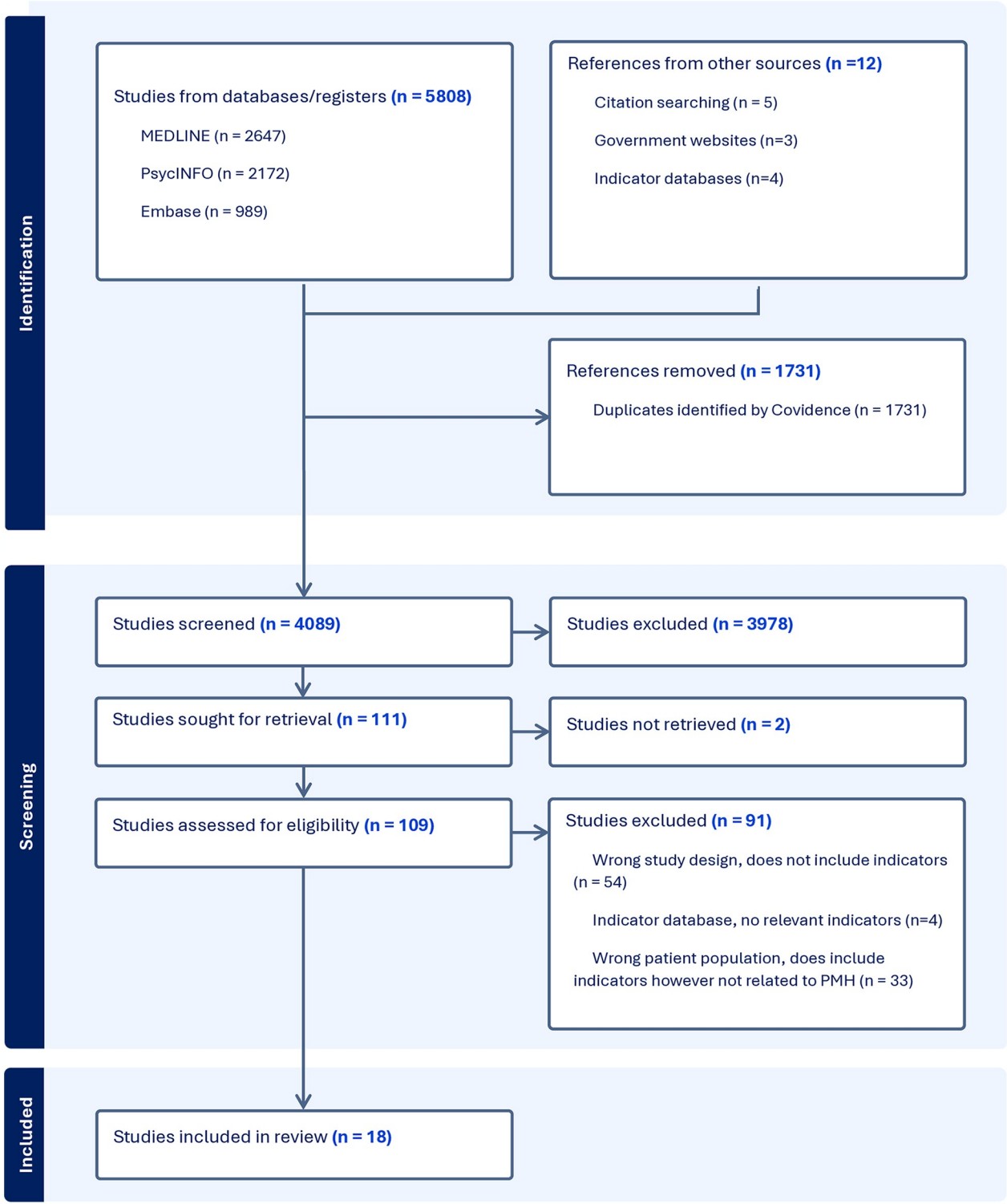

**Fig 1. PRISMA flow diagram.**

Measurement developed in the UK, which includes an indicator for postpartum depression. They found significant gaps in the mental health knowledge of their healthcare staff, which required specific training to implement the resource. They also found no women reporting a prior diagnosis of a PMH disorder despite many reporting symptoms [35]. This highlights the importance of global indicators being universally valid.

The WHO recommends that PMH care be integrated into existing maternal and child health services [18]. Whilst this is an important step in improving accessibility of PMH care, additional steps are necessary in advancing towards the goal of universal high-quality PMH care for all perinatal women. A cornerstone of indicator implementation involves the ability of health systems and data collection platforms to monitor progress. Currently, however, there are significant limitations to implementing a core set of PMH indicators as required data is not routinely captured in many countries around the world. A further important step is the alignment of all relevant guidelines to include PMH, this would include updating the WHO recommendations on antenatal care for a positive pregnancy experience to include PMH.

This scoping review has several limitations. The indicator data available was heterogenous and often incomplete. For example, many resources had indicators with incomplete definitions, or excluded the numerator, denominator, or possible data source. The definitions of each indicator also varied across sources, with differing use of terminologies when describing mental health disorders. An inability to populate all indicators and generalisation of some definitions to form a universal set of indicators may limit the feasibility of implementation. The scope of this research also excluded risk factors and social determinants for PMH, which may have addressed gaps in measurement for the other PMH ToC domains. The grey search of existing indicator databases and webpages was in English only, which could have led to an exclusion of indicators from LMICs. However, a lack of PMH indictors in both LMICs and HICs was well-recognised by experts consulted during this review, highlighting the urgent need for further research in this area. We also imposed a time limit on the papers selected (from January 2000). We felt that the last 24 years was an adequate time frame to include contemporary studies, but it is possible that we omitted studies published before this time.

## Conclusion

The identified PMH indicators in this review provide an account of the current landscape of PMH measurement globally and is the first step in the development of a monitoring framework by the WHO to be used by countries to monitor progress and make programme adjustments. Future research would include the development of new indicators which address the gaps in the PMH ToC interpersonal relationship, community, and policy landscape domains. The development and adaptation of indicators in LMICs would also be vital for future implementation in these settings. Important next steps in this work include further consultation with stakeholders to prioritise and operationalise a core set of agreed indicators, and technical assessment of data sources to determine the feasibility of implementing the indicator set.

This review forms part of the foundational work for the development of a global monitoring framework which will monitor progress towards the provision of universal high quality PMH care. This has implications at a country and global level for the measurement and monitoring of PMH, an important component of the Sustainable Development Goals.

## Supporting information

**S1 Appendix. Scoping review protocol.**
(DOCX)

**S1 Table. Preferred Reporting Items for Systematic reviews and Meta-Analyses extension for Scoping Reviews [PRISMA-ScR] checklist.**
(DOCX)

**S2 Table. Search strategy.**
(DOCX)

**S3 Table. Indicator data from each resource.**
(DOCX)

## Acknowledgments

We would like to thank the following groups for their feedback and input during the consultation process of this review:

WHO Department of Maternal, Newborn, Child, Adolescent Health and Ageing.

WHO Mother and Newborn Information Tracking Outcomes and Results group.

WHO Department of Mental Health and Substance Use.

USAID MOMENTUM Perinatal Mental Health Community of Practice members.

## Author Contributions

**Conceptualization:** Alexandra Roddy Mitchell, Elissa Kennedy, Allisyn C. Moran, Francesca Palestra, Caroline S. E. Homer.

**Data curation:** Elly Layton, Alexandra Roddy Mitchell.

**Formal analysis:** Elly Layton, Alexandra Roddy Mitchell.

**Funding acquisition:** Elissa Kennedy, Caroline S. E. Homer.

**Investigation:** Elly Layton.

**Methodology:** Elly Layton, Allisyn C. Moran, Francesca Palestra, Shanon McNab, Caroline S. E. Homer.

**Project administration:** Elly Layton.

**Supervision:** Elissa Kennedy, Caroline S. E. Homer.

**Validation:** Elly Layton, Allisyn C. Moran, Francesca Palestra, Neerja Chowdhary, Shanon McNab.

**Visualization:** Elly Layton.

**Writing – original draft:** Elly Layton, Alexandra Roddy Mitchell.

**Writing – review & editing:** Alexandra Roddy Mitchell, Elissa Kennedy, Allisyn C. Moran, Francesca Palestra, Neerja Chowdhary, Shanon McNab, Caroline S. E. Homer.

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
