## [Decision Letter · Decision Letter 0]

21 Nov 2024

PONE-D-24-46366Maternal Mental Health Matters: A scoping review of indicators for perinatal mental healthPLOS ONE

Dear Dr. Homer,

Thank you for submitting your manuscript to PLOS ONE. After careful consideration, we feel that it has merit but does not fully meet PLOS ONE’s publication criteria as it currently stands. Therefore, we invite you to submit a revised version of the manuscript that addresses the points raised during the review process. Please submit your revised manuscript by  Jan 05 2025 11:59PM. If you will need more time than this to complete your revisions, please reply to this message or contact the journal office at plosone@plos.org. Please include the following items when submitting your revised manuscript:A rebuttal letter that responds to each point raised by the academic editor and reviewer(s). You should upload this letter as a separate file labeled 'Response to Reviewers'.A marked-up copy of your manuscript that highlights changes made to the original version. You should upload this as a separate file labeled 'Revised Manuscript with Track Changes'.An unmarked version of your revised paper without tracked changes. You should upload this as a separate file labeled 'Manuscript'.

We look forward to receiving your revised manuscript.

Kind regards,

Maria José Nogueira, Ph.D.

Academic Editor

PLOS ONE

Journal Requirements:

2. Please expand the acronym WHO (as indicated in your financial disclosure) so that it states the name of your funders in full.

3. Thank you for stating the following in your Competing Interests section: [No].

5. We notice that your supplementary files are included in the manuscript file. Please remove them and upload them with the file type 'Supporting Information'. Please ensure that each Supporting Information file has a legend listed in the manuscript after the references list.

Reviewers' comments:

Reviewer's Responses to Questions

**Comments to the Author**

1. Is the manuscript technically sound, and do the data support the conclusions?

Reviewer #1: Partly

Reviewer #2: Yes

2. Has the statistical analysis been performed appropriately and rigorously? 

Reviewer #1: N/A

Reviewer #2: N/A

3. Have the authors made all data underlying the findings in their manuscript fully available?

Reviewer #1: Yes

Reviewer #2: Yes

4. Is the manuscript presented in an intelligible fashion and written in standard English?

Reviewer #1: Yes

Reviewer #2: Yes

5. Review Comments to the Author

Reviewer #1: Very pertinent study, with good theoretical foundation.

The title of an SR must always end with: A Scoping Review and not in the way it is presented. The abstract must contain a set of information that is not clear in the study. They mention that they carry out an SR but do not follow the principles of their research protocol and the recommendation (Preferred Reporting Items for Systematic reviews and Meta-Analyses extension for Scoping Reviews (PRISMA-ScR) Checklist). Do not put acronyms on abstract. Unclear method. It is recommended that the SR protocol is registered and included in the Open Science Framework (OSF), for example, before publishing the report. They should not have chronological lines to map out all the evidence on the topic under study. It would have been easier if they had included the gray literature they included in the research in the diagram. It is confusing for anyone who reads how you did the research and how you arrived at the results. When extracting the results, I would like to have seen the PRISMA flow diagram explained. From what they seem, it is not clear how they proceeded with data extraction and analysis. Dot provides numbers of evidence sources selected, assessed for eligibility and included in the review, with reasons for exclusions at each stage, using the flow diagram. Removal of results is unclear. Very confusing. Do not state the process for selecting sources of evidence (i.e., screening and eligibility) included in the scoping review. In the "Summary of indicators mapped against PMH ToC" and the "Supplementary file 3. Indicator data extracted from each resource", there are incorrect references to all resources. in the text it presents one type of references and in the table another. must be uniform in accordance with the magazine's requests. Korst et al (US) no references nº (Table1). It would be interesting to provide a general interpretation of the results in relation to the review questions and objectives, as well as possible implications. There are acronyms that are not spelled out, for example CDC.

Reviewer #2: Mertodologia:

A search was carried out (Medline, Embase PsycInfo ). Why didn't you search Pubmed, the largest health database?

The term perinatal period | No results found as a DeCS/MeSH term, Health Sciences Descriptors

The terms mental health and indicator correspond to several DeCS/MeSH and Health Sciences Descriptors, so it's essential to include the ID to identify which one was used in the search.

Supplementary file one does not mention which Boolean phrases were used in the search, so we could not replicate the search.

We also searched Governmental websites from English-speaking countries; it would be essential to mention the phrase in natural language as they did that search.

In the Prisma flowchart, it would be necessary to mention whether or not there were findings in this way and through direct contact with researchers (grey literature).

The description of the results should include the overall number of articles found or the PRISMA flowchart to make it easier to understand.

Very confusing discussion:

They present the limitations of the review in the discussion and should emphasise them at the end of the article.

Including a short conclusion withgs and implications for practice would be essential the main findin.

6. PLOS authors have the option to publish the peer review history of their article (what does this mean?). If published, this will include your full peer review and any attached files.

Reviewer #1: **Yes: **Sara Palma

Reviewer #2: **Yes: **Delfina Ana Pereira Ramos Teixeira

---

## [Author Response · Author response to Decision Letter 0]

20 Dec 2024

5. Review Comments to the Author

Reviewer #1: Very pertinent study, with good theoretical foundation.

The title of an SR must always end with: A Scoping Review and not in the way it is presented. 

Thank you for your feedback. We have changed the title accordingly to now read: “Maternal Mental Health Matters: Indicators for Perinatal Mental Health - : A Scoping Review”

The abstract must contain a set of information that is not clear in the study. They mention that they carry out an SR but do not follow the principles of their research protocol and the recommendation (Preferred Reporting Items for Systematic reviews and Meta-Analyses extension for Scoping Reviews (PRISMA-ScR) Checklist). 

We have edited the Abstract to provide the key information more clearly. We have followed the principles of a scoping review. This is detailed in the PRIMSA-ScR checklist which is now found in S1.

Do not put acronyms on abstract. There are acronyms that are not spelled out, for example CDC.

This has been addressed. We have ensured all acronyms have been spelled out.

It is recommended that the SR protocol is registered and included in the Open Science Framework (OSF), for example, before publishing the report. 

The scoping review is not registered, however our protocol has now been added as a supplementary file (S1 Appendix). 

They should not have chronological lines to map out all the evidence on the topic under study. 

We are not clear as to what this comment refers to, so we have been unable to respond. We are unclear of the meaning of this comment, if further clarification could be provided, we would be happy to address it. 

It would have been easier if they had included the gray literature they included in the research in the diagram. 

We have also now included grey literature in the PRISMA flow diagram (Fig 1) and included an exclusion criterion in the body of the manuscript. Rationale for choice of grey literature has also been included.

It is confusing for anyone who reads how you did the research and how you arrived at the results. When extracting the results, I would like to have seen the PRISMA flow diagram explained. From what they seem, it is not clear how they proceeded with data extraction and analysis. Dot provides numbers of evidence sources selected, assessed for eligibility and included in the review, with reasons for exclusions at each stage, using the flow diagram. 

Thank you for this feedback. We have now explained the PRISMA flow diagram in the results section and included it as Figure 1 (opposed to as a supplementary file). Results section (line 150) now reads: 

“A total of 5808 resources from three databases were retrieved (Figure 1). An additional 12 resources from other sources including government websites and international agency websites were also retrieved. After duplicates were removed 4082 records were screened at the title/abstract stage and 109 at full text. Overall, 2 studies could not be retrieved and 91 were excluded at full text screening, resulting in 18 resources for inclusion.”

Removal of results is unclear. Very confusing. Do not state the process for selecting sources of evidence (i.e., screening and eligibility) included in the scoping review. In the "Summary of indicators mapped against PMH ToC" and the "Supplementary file 3. 

We have clarified this and included more about what we did in the Methods and the results of this are now in the Findings. We hope that makes it clearer.

Indicator data extracted from each resource", there are incorrect references to all resources. in the text it presents one type of references and in the table another. must be uniform in accordance with the magazine's requests. Korst et al (US) no references nº (Table1). 

References have been updated to be consistent throughout the text and tables.

It would be interesting to provide a general interpretation of the results in relation to the review questions and objectives, as well as possible implications. 

We have revised the Discussion to address this comment. The possible implications have also been clarified in the discussion.

Reviewer #2: Mertodologia:

A search was carried out (Medline, Embase PsycInfo ). Why didn't you search Pubmed, the largest health database?

We did not search the PubMed interface as we searched the Medline database directly through the Ovid interface.

The term perinatal period | No results found as a DeCS/MeSH term, Health Sciences Descriptors

S2 Table includes all search terms used. The term perinatal was used however pregnancy, postnatal, peripartum, etc were also used. This was done with the aim of capturing all relevant resources even if some were specific to pregnancy or postpartum, but therefore still meet the definition of the perinatal period.

The terms mental health and indicator correspond to several DeCS/MeSH and Health Sciences Descriptors, so it's essential to include the ID to identify which one was used in the search.

Supplementary file one does not mention which Boolean phrases were used in the search, so we could not replicate the search.

We have added a note to the supplementary table which now describes which search terms were MeSH terms/keywords and the search should therefore be reproducible.

We also searched Governmental websites from English-speaking countries; it would be essential to mention the phrase in natural language as they did that search.

We have now included more information on how we searched governmental websites in the methods. Methods section (line 112) now reads: “For each of these resources, available indicator dashboards were systemically reviewed for relevant indicators and the keyword ‘perinatal mental health’ was searched.”

In the Prisma flowchart, it would be necessary to mention whether or not there were findings in this way and through direct contact with researchers (grey literature).

The methods have also been updated to expand the search strategy of the grey resources. These have also been included in the PRISMA (Fig 1). We did not find any extra resources through contact with researchers, which we have now stated in the methods.

The description of the results should include the overall number of articles found or the PRISMA flowchart to make it easier to understand.

Thank you for this feedback. We have now explained the PRISMA flow diagram in the results section and included it as Figure 1 (opposed to as a supplementary file). Results section (line 150) now reads: 

“A total of 5808 resources from three databases were retrieved (Figure 1). An additional 12 resources from other sources including government websites and international agency websites were also retrieved. After duplicates were removed 4082 records were screened at the title/abstract stage and 109 at full text. Overall, 2 studies could not be retrieved and 91 were excluded at full text screening, resulting in 18 resources for inclusion.”

Very confusing discussion:

They present the limitations of the review in the discussion and should emphasise them at the end of the article.

Including a short conclusion withgs and implications for practice would be essential the main findin.

We have worked on the Discussion to improve the flow. We have clarified the implications of this work in the discussion.

---

## [Decision Letter · Decision Letter 1]

6 Jan 2025

PONE-D-24-46366R1Maternal Mental Health Matters: Indicators for Perinatal Mental Health - A Scoping ReviewPLOS ONE

Dear Dr. Homer, 

Thank you for submitting your manuscript to PLOS ONE. After careful consideration, we feel that it has merit but does not fully meet PLOS ONE’s publication criteria as it currently stands. Therefore, we invite you to submit a revised version of the manuscript that addresses the points raised during the review process.

We look forward to receiving your revised manuscript.

Kind regards,

Maria José Nogueira, Ph.D.

Academic Editor

PLOS ONE

Journal Requirements:

Reviewers' comments:

Reviewer's Responses to Questions

**Comments to the Author**

1. If the authors have adequately addressed your comments raised in a previous round of review and you feel that this manuscript is now acceptable for publication, you may indicate that here to bypass the “Comments to the Author” section, enter your conflict of interest statement in the “Confidential to Editor” section, and submit your "Accept" recommendation.

Reviewer #1: All comments have been addressed

Reviewer #2: All comments have been addressed

2. Is the manuscript technically sound, and do the data support the conclusions?

Reviewer #1: Yes

Reviewer #2: Yes

3. Has the statistical analysis been performed appropriately and rigorously? 

Reviewer #1: N/A

Reviewer #2: N/A

4. Have the authors made all data underlying the findings in their manuscript fully available?

Reviewer #1: Yes

Reviewer #2: Yes

5. Is the manuscript presented in an intelligible fashion and written in standard English?

Reviewer #1: Yes

Reviewer #2: Yes

6. Review Comments to the Author

Reviewer #1: Thank you very much for your concern in making the necessary rectifications.

Regarding the time limits for research. The authors selected the publication date limit from 1 January 2000 to 4 October 2024. I question why they selected this period, since ScR should not have a period for research?

However, i suggested, what they show in "data extraction and analysis", lines 135-140, must be presented in the results and what they show in lines 143-147 in "data extraction and analysis".

Lastly, in the discussion of the results, there is no comparison of authors. I think it would enrich the results.

Reviewer #2: The authors present the methodology using the PRIMA flowchart and complement it with annexes, making the results much easier to understand.

The way they present the results of the articles also makes them easier to read and understand, as does the discussion.

However, they could separate the conclusion or a summary and the limitations of discussing the results.

7. PLOS authors have the option to publish the peer review history of their article (what does this mean?). If published, this will include your full peer review and any attached files.

Reviewer #1: **Yes: **Sara Elisabete Cavaco Palma

Reviewer #2: **Yes: **Delfina Ana Pereira Ramos Teixeira

---

## [Author Response · Author response to Decision Letter 1]

6 Jan 2025

Thank you for the feedback regarding: Maternal Mental Health Matters: Indicators for Perinatal Mental Health - A Scoping Review.

Our response to each point from the reviewers can be found below

Reviewer #1: Thank you very much for your concern in making the necessary rectifications.

Regarding the time limits for research. The authors selected the publication date limit from 1 January 2000 to 4 October 2024. I question why they selected this period, since ScR should not have a period for research?

We wanted to include recent research and felt that the year 2000 was as far back as we wanted to go. We have added that justification in the Methods and also a line under limitations. 

However, i suggested, what they show in "data extraction and analysis", lines 135-140, must be presented in the results and what they show in lines 143-147 in "data extraction and analysis".

Lines 135-140 are the section under Data extraction and analysis. This is clearly the process which was undertaken and not the findings so we would prefer to keep it in the Methods as is commonly presented this way. 

Lines 143-147 do present the findings – the results of the search process. 

Lastly, in the discussion of the results, there is no comparison of authors. I think it would enrich the results.

We are not really sure what is meant here. We have never presented a comparison of authors before in a scoping review. The authors are all listed in the supplementary files. 

Reviewer #2: The authors present the methodology using the PRIMA flowchart and complement it with annexes, making the results much easier to understand.

The way they present the results of the articles also makes them easier to read and understand, as does the discussion.

However, they could separate the conclusion or a summary and the limitations of discussing the results.

There is a paragraph on the limitations already within the Discussion.

We have made a heading for the Conclusion as suggested.

---

## [Editor Report · Decision Letter 2]

9 Jan 2025

Maternal Mental Health Matters: Indicators for Perinatal Mental Health - A Scoping Review

PONE-D-24-46366R2

Dear Dr. Caroline SE Homer

We’re pleased to inform you that your manuscript has been judged scientifically suitable for publication and will be formally accepted for publication once it meets all outstanding technical requirements.

Kind regards,

Maria José Nogueira, Ph.D.

Academic Editor

PLOS ONE
---

## [Editor Report · Acceptance letter]

16 Jan 2025

PONE-D-24-46366R2 

PLOS ONE

Dear Dr. Homer, 

I'm pleased to inform you that your manuscript has been deemed suitable for publication in PLOS ONE. Congratulations! Your manuscript is now being handed over to our production team.

Kind regards, 

on behalf of

Professor Maria José Nogueira 

Academic Editor

PLOS ONE